# Controlling cyclodextrin host-guest complexation in water with dynamic pericyclic chemistry

Marius Gaedke [1], Anja Ramström [1], Daisy R. S. Pooler [1] & Fredrik Schaufelberger [1,2] ✉

Water-soluble macrocycles are useful molecular hosts for drug delivery, stimuli-responsive materials, water purification and many other applications. However, controlling the host-guest chemistry of macrocycles such as cyclodextrins under physiologically relevant conditions is a major challenge. Here we demonstrate the use of dynamic pericyclic chemistry to derivatise guests for cyclodextrins under mild conditions, thereby turning off molecular recognition. We show that the Diels-Alder [4 + 2] cycloaddition reaction between anthracene derivatives and activated alkenes proceed rapidly, selectively and reversibly in water under ambient conditions. This reaction can be used to modulate binding of both native and modified β-cyclodextrins to the anthracene. By appropriate choice of conditions, the resulting chemical reaction network could also operate under non-equilibrium steady state conditions. Finally, alkene scavengers could induce the retro-Diels Alder reactions, allowing the use of the pericyclic reaction system as a molecular switch.

Cyclodextrins (CDs) are cyclic oligosaccharides extensively used for drug delivery, water purification, advanced materials, analytical separations or directly as therapeutic treatments[1–5]. Key to these applications is the ability of CDs to form inclusion complexes with hydrophobic guests in both solution and the solid state. Encapsulation leads to large changes in the physicochemical properties of guest molecules, and affects their biological and pharmaceutical behaviour.

By controlling CD binding, one can achieve modified or prolonged drug release profiles[6], stimuli-responsive behaviour in materials[7,8] or control over motion in artificial molecular machinery[9–11]. Despite the importance of cyclodextrin complexation, the number of methods for user-controlled guest binding is limited[12–16]. While CD host–guest chemistry can be controlled using light, pH or redox stimuli, there are few methods for controlled binding using selective chemical reactions[17,18]. This is surprising, as chemical stimuli are complementary to exogenous input such as light and can be useful to detect biomarkers or other internal disease factors.

Chemical stimuli for drug delivery should preferentially be water-soluble, biocompatible and induce a reversible response. Inspired by the use of pericyclic reactions in biorthogonal chemistry[19] and dynamic covalent chemistry for assembly of mechanically interlocked molecules[20–23], we hypothesised that dynamic pericyclic reactions could be useful for controlling host–guest binding in water.

The field of dynamic covalent chemistry is currently thriving[24–28], but dynamic pericyclic reactions with reversibility under mild conditions is still

an unmet challenge[29–31]. Most pericyclic reactions require harsh conditions for reversibility (i.e. T > 150 °C), using reagents with poor solubility in water. One of the few systems of broader use is the reversible [4 + 2] Diels–Alder cycloaddition reaction between activated alkenes and anthracene derivatives, which was explored by Lehn in organic solvents[32,33]. Since anthracenes bind with moderate affinity to cyclodextrins in water[34], we reasoned that a sufficiently selective dynamic pericyclic reaction could deactivate the CD binding to anthracene by virtue of increased steric bulk. This would liberate free CD into solution, where the macrocycle could perform other functions of interest[35–38].

Here we show the realisation of such an approach for controlled host–guest binding in water (Fig. 1a). Several activated dienes and anthracenes are compatible with this reaction system, and the reaction works in pure water in a fast, reversible and selective manner at ambient temperature. By manipulating the equilibrium via the introduction of alkene scavengers, we could also introduce more advanced functionality into the system, including non-equilibrium steady-state behaviour[39,40] and bistate switching[41–47].

## Results and discussion
### Synthesis and binding studies

Initially, we studied the binding between cyclodextrins and 2, 6-dioxyanthracene scaffolds in water (Fig. 1b). Anthracenes with this substitution pattern are readily synthesised from anthraflavic acid by simple

[1]Department of Chemistry, KTH Royal Institute of Technology, Stockholm, Sweden. [2]Department of Chemistry, University of Warwick, Coventry, UK.
✉e-mail: fredrik.schaufelberger@warwick.ac.uk

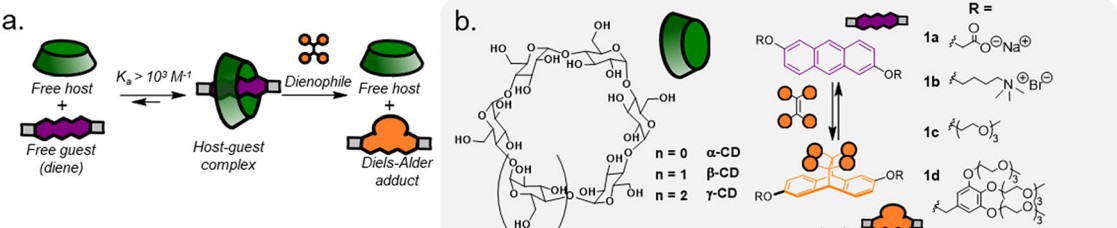

**Fig. 1 | Control of host–guest chemistry by pericyclic reactions. a** Schematic of dynamic pericyclic reaction control of cyclodextrin host–guest binding. **b** Chemical structures of the core compounds in this work, as well as illustration of the [4 + 2]cycloaddition reaction of anthracenes **1a–1d** with activated alkenes.

**Fig. 2 | Titration data showing β-CD-anthracene binding.** $^1$H-NMR spectroscopic titration (400 MHz, D$_2$O, 298 K) of increasing equivalents of β-CD (green boxes) added into a 2 mM solution of **1a**.

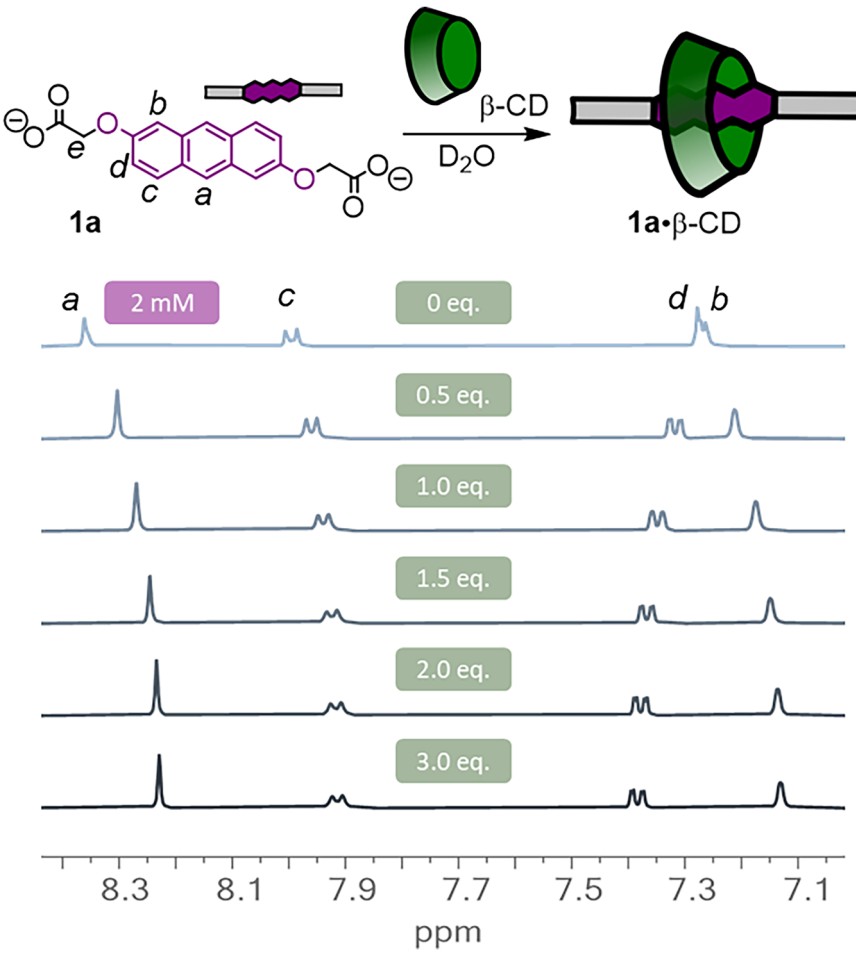

reduction with NaBH$_4$. Subsequent difunctionalisation to install solubilising groups (Supplementary Section S3) was straightforward and led to molecules with negative charges (carboxylates, **1a**), positive charges (quaternary ammonium ions, **1b**) and neutral triethylene glycol units (TEG, **1c** and **1d**). Unlike previous reports of dynamic pericyclic reactions with anthracenes[32,33,48–51], we kept the central 9,10-positions of the anthracenes unsubstituted to promote binding to macrocyclic hosts. This decreases the oxidative stability of the molecules, yet we found that once dialkylated, this class of molecules were bench-stable in the absence of light.

Anthracenes form inclusion complexes with for example cycloparaquats, cucurbiturils and CD macrocycles[34,52,53]. To exploit this feature, we explored the binding of anthracenes **1a–1d** to a range of cyclodextrin macrocycles. Titration of the cyclodextrin into a solution of the anthracene in D$_2$O at ambient temperature was followed by $^1$H-NMR spectroscopy, from which the complexation-induced binding shifts could be analysed using a 1:1 binding model and the BindFit software (Fig. 2)[54]. Most processes

were in fast exchange, and the corresponding binding constants are summarised in Table 1 (see further Supplementary Figs. S8–S14). α-CD was too small to bind to anthracene **1a**, in line with literature precedence (entry 1)[34]. Pleasingly, β-CD and anthracene **1a** showed proficient 1:1 binding with a binding constant of $3.6 \times 10^3$ M$^{-1}$ (entry 2). The larger γ-CD led to more complicated binding where simultaneous 2:1 and 1:1 guest binding was observed, with a mix between fast (for 2:1) and slow (for 1:1) exchange regimes (entry 3 and Supplementary Fig. S12). Given the complexity of this system, we hence focused on β-CD for all subsequent studies.

The positively charged anthracene **1b** also displayed moderate binding to β-CD at $7.4 \times 10^3$ M$^{-1}$ (entry 4). Neutral TEG-functionalised anthracene **1c** was however not sufficiently soluble in D$_2$O for accurate binding determination, even in the presence of solubilising β-CD. Finally, the fully water-soluble more heavily TEG-substitute anthracene **1d** did not bind at all to β-CD (entries 5–6). Even extended heating at 70 °C to accelerate threading kinetics did not lead to complexation of **1d**, despite preliminary

## Table 1 | Equilibrium constants for host–guest complexation between anthracenes and CDs (D₂O, RT, 2 mM anthracene guest)

| Entry | Host | Guest | $K_a$ (M$^{-1}$) |
|---|---|---|---|
| 1 | α-CD | 1a | – |
| 2 | β-CD | 1a | $3.6 \pm 0.079 \times 10^3$ |
| 3 | γ-CD | 1a | Non-trivial binding |
| 4 | β-CD | 1b | $7.4 \pm 1.5 \times 10^3$ |
| 5 | β-CD | 1c | – |
| 6 | β-CD | 1d | – |
| 7 | β-CD-OMe$_{(10–12)}$ | 1a | $2.0 \pm 0.092 \times 10^3$ |
| 8 | β-CD-N₃ | 1a | $7.8 \pm 0.45 \times 10^3$ |

Four equiv. NaOD was added when guest 1a was used. Binding constants were estimated using a 1:1 model and the BindFit software at http://supramolecular.org.

## Table 2 | Forward Diels–Alker reaction between anthracene 1a and different dienophiles[a]

| Entry | Dienophile | $t_{1/2}$ | $k$ (M$^{-1}$s$^{-1}$) |
|---|---|---|---|
| 1 | 2a | 28d | $2.9 \times 10^{-5}$ |
| 2 | 2b | n.r. | n.r. |
| 3 | 2c | n.r. | n.r. |
| 4 | 2d | Decomposition | – |
| 5 | 2e | 10 h | N/A[b] |
| 6 | 2f | <1 min | $>10^3$ |
| 7 | 2g | 9 h | $2.0 \times 10^{-3}$ |
| 8 | 2h | Decomposition | – |
| 9 | 2i | n.r. | n.r. |

*n.r.* no reaction.
[a]Reaction conditions: D₂O with 4 equiv. NaOD, 1a (1.0 mM, 1 equiv.), dienophile (10 equiv.), RT.
[b]Heterogenous reaction mixture.

molecular modelling indicating β-CD should be capable to thread over the phenolic substituents (see Supporting Information Section S9 for further details). Finally, derivatised β-cyclodextrins (methylated, azide-substituted, entries 7–8) bound to anionic anthracene **1a** with similar strength, confirming that functionalised CDs are compatible with this system.

### Diels–Alder reaction studies

We next studied the Diels–Alder reaction in water between 2,6-dioxyanthracenes and activated alkenes and alkynes (Table 2). Satisfyingly, we found several dienophiles which underwent complete Diels–Alder reactions with dicarboxylate anthracene **1a** in D₂O (1 mM) at room temperature (see Supplementary Table S1 for comparison data in CDCl₃, and for additional substrates without reactivity). Diethyl fumarate **2a** gave the desired Diels–Alder product **1a2a**, albeit with slow reaction rate (entry 1) (Conversion to Diels-Alder adducts was determined by observing the ¹H-NMR singlet shift of the 9,10-protons on the anthracene scaffold, which generally moved from ~8.4 ppm to <7 ppm.). The *cis*-isomer of **2a**, diethyl maleate **2b**, unexpectedly lacked reactivity under these conditions, as did the structurally related maleic acid **2c** (entries 2–3). The reason for the failure of these substrates to react is currently unclear to us. Maleic anhydride **2d** typically undergoes facile Diels–Alder reactions and is a competent dienophile in organic solvents, however under our aqueous reaction conditions the hydrolysis process outcompeted the desired pericyclic reaction (entry 4).

To improve the reaction kinetics, we tested alkene **2e**, decorated with four rather than two electron withdrawing groups. As expected, the reaction rate improved considerably, but the interpretation of the results was complicated by the limited solubility (entry 5). Moving to the even more electron-poor tetracyanoethylene (TCNE) **2f** (which was fully soluble under the reaction conditions), complete conversion to the Diels–Alder adduct **1a2f** was observed within one minute (entry 6).

We further tested a series of dienophiles used in the field of biorthogonal chemistry[19]. Pleasingly, we found that maleimides – a generally well-tolerated functional group for biological applications – underwent fast Diels–Alder cycloadditions. Compound **2g** reacted cleanly with anthracene **1a** to yield the adduct **1a2g** in 24 h (entry 7). Dibenzocyclooctynes, such as **2h**, are used for biorthogonal strain-promoted alkyne-azide cycloadditions[19]. However, **2h** did not react with **1a** under these conditions, and only decomposition was observed (entry 8). Similarly, linear alkynes also did not engage in this chemistry at ambient temperatures (entry 9).

Fortunately, both TCNE **2f** and maleimide **2g** reacted in a similar manner with all anthracenes regardless of the solubilising group, confirming that the observed reactivity is general (Note that **1c** is only sparingly soluble in water, so Diels-Alder reactions were instead tested in CDCl₃ and DMSO-*d₆* for this compound.). Isolation of the Diels–Alder adducts was complicated by rapid reversibility during all attempts at chromatographic purification, a phenomenon also observed with other dynamic covalent bonds[24]. However, for **1a** and **1b**, analysis using in situ ¹H-NMR spectroscopy confirmed full conversion as indicated by the large NMR shifts (Δδ > 2 ppm) of the 9,10-anthracene protons upon rehybridisation from sp² to sp³ (Supplementary Figs. S15–S18). High resolution mass spectrometry (HRMS) further confirmed conversion into the expected Diels–Alder adducts (Supplementary Figs. S21–S24).

Diels–Alder reactions in water often proceed faster than in organic solvents, and a number of explanations for this phenomena – such as hydrogen-bond activation and the hydrophobic effect – have been put forward[55–57]. As pericyclic reactions with anthracene in water are underexplored, we wanted to investigate the kinetics and activation parameters of the reaction in more detail. A full Eyring analysis for the model reaction of carboxylate-adorned anthracene **1a** with maleimide dienophile **2g** to form Diels–Alder adduct **1a2g** was hence conducted (Fig. 3, Supplementary Figs. S1–S3). Kinetic measurements with ¹H-NMR spectroscopy between 6 and 40 °C allowed determination of the activation parameters as $\Delta H^{\ddagger} = +11.6$ kcal mol$^{-1}$ and $\Delta S^{\ddagger} = -39.6$ cal K$^{-1}$ mol$^{-1}$, with an overall barrier of $\Delta G^{\ddagger} = 23.2$ kcal mol$^{-1}$ at 293 K. These results are in close agreement with previously reported activation parameters for Diels–Alder reactions in organic solvents[48–51]. Therefore, the aqueous environment does not seem to significantly affect the kinetics of the reaction.

### Host–guest control via Diels–Alder reaction

Next, we tested our core hypothesis of directly disrupting the CD-anthracene host–guest structure via a dynamic pericyclic reaction (Fig. 1). We prepared the host–guest complex **1a•β-CD** by adding β-CD (1 equiv.) into a 1 mM solution of **1a** in D₂O. Subsequently, TCNE **2f** (20 equiv.) was added in one batch. Analysis via ¹H-NMR spectroscopy confirmed the

**Fig. 3 | Time-dependent conversion of Diels–Alder reactions with anthracene 1a.** Conversion over time for the Diels–Alder reaction of **1a** (2mM) and **2g** (10 equiv.) at different temperatures, as measured by ¹H-NMR spectroscopy (D₂O, 8 mM NaOD).

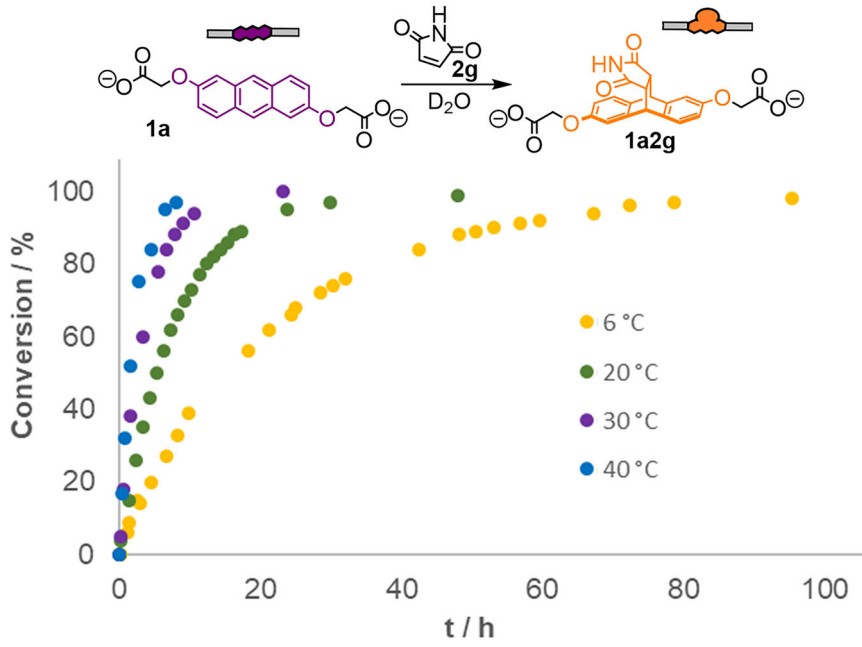

**Fig. 4 | Guest displacement from β-CD host due to a selective Diels–Alder reaction. a** Truncated ¹H NMR spectra (400 MHz, D₂O, 298 K) of (from top to bottom) **1a2f** and free β-CD, **1a**•β-CD, **1a** and β-CD. Initial concentration of

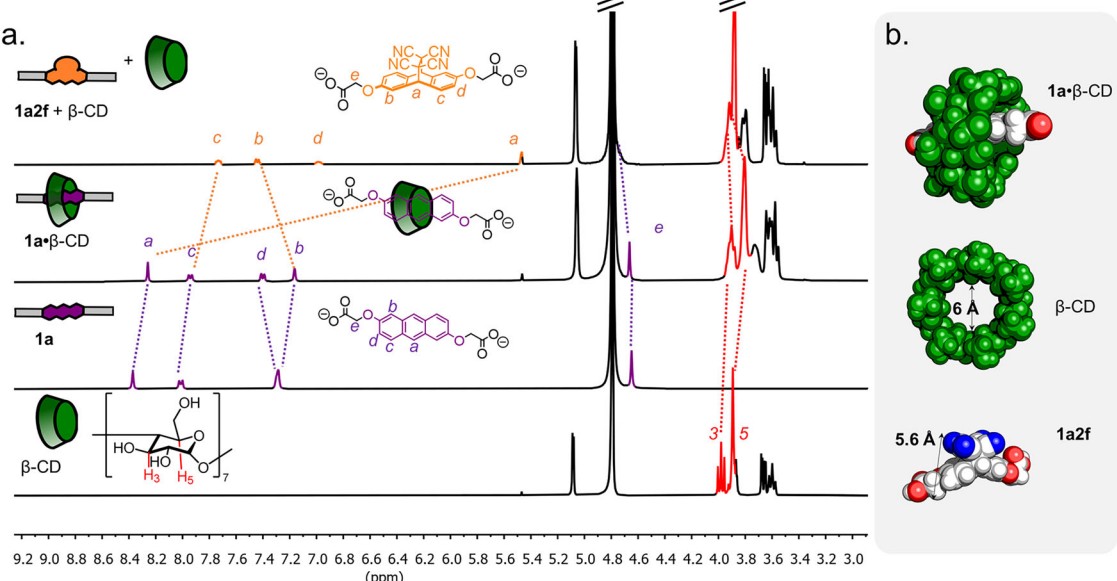

**1a** = 1 mM. **b** Molecular models of **1a**•β-CD, β-CD and **1a2f**, calculated at the GFN2-xTB/AL PB(H₂O) level of theory. Relevant distance measurements indicated with arrows.

transformation of anionic anthracene **1a** into Diels–Alder adduct **1a2f** (Fig. 4a), as indicated by large shifts of aromatic anthracene protons *a-d*. More significantly, analysis of β-CD protons *3* and *5* (i.e. the axial glycosidic 3- and 5-positions) confirmed expulsion of the guest from the β-CD cavity through typical downfield shifts ($\Delta\delta = 0.1$ ppm) after addition of TCNE **2f**. This observation confirms that the Diels–Alder reaction indeed led to modulation of the host–guest binding. The same experiment was repeated with cationic anthracene **1b**, providing an identical outcome (Supplementary Fig. S25).

In theory, the Diels–Alder reaction can occur either inside the CD cavity or on unbound anthracene free in solution. However, we observed that there was no noticeable change in the kinetics of the Diels–Alder reaction of **1a** with either **2f** or **2g** in the presence or absence of β-CD (22 equiv). This indicates that the reaction can still take place on the

encapsulated anthracene, but that there seems to be no major rate accelerations as a result of potential co-inclusion or similar.

We further corroborated these results through semi-empirical quantum mechanics calculations, using the Conformer Rotamer Ensemble Sampling Tool at the GFN2-xTB level with implicit H₂O solvation (ALPB solvent model)[58,59]. For a 1:1 host–guest complex **1a**•β-CD, the located energy minimum positioned the guest inside the host in a pseudo[2] rotaxane manner (Fig. 4b). Upon reaction, the anthracene 9,10-carbons undergo rehybridisation from sp² to sp³, meaning the previously planar structure becomes bent, with the dienophile addition further increasing steric bulk. The optimised structures (GFN2-xTB/ALPB(H₂O)) of the TCNE-derived Diels–Alder product **1a2f** as well as free β-CD host reveals relatively similar spatial dimensions (Fig. 4b), as the cavity diameter (ca. 6 Å) is roughly equal to the shortest transversal measure for **1a2f** (5.6 Å). While

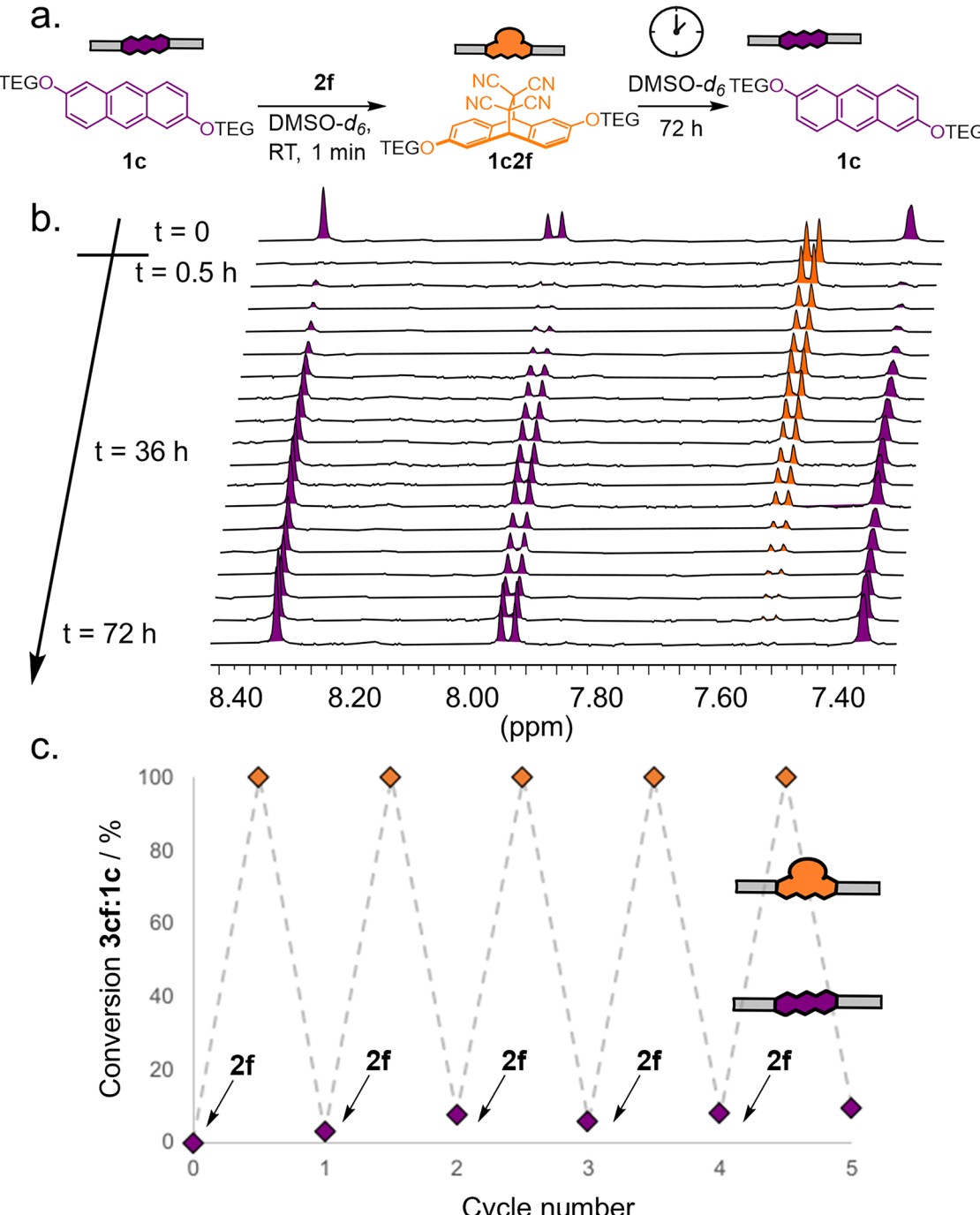

**Fig. 5 | Transient conversion to pericyclic reaction products using DMSO-$d_6$ as an alkene scavenger. a** Scheme of transient pericyclic reaction of **1c** and **2f** to form **1c2f** in DMSO-$d_6$. **b** Truncated $^1$H-NMR spectra (400 MHz, DMSO-$d_6$, 298 K) showing time course evolution of **1c** (1 mM, top spectra), its complete transformation into **1c2f** upon **2f** addition (second spectra from top) and its spontaneous evolution back to **1c** over 72 h (spectra 3 to bottom). **c** Fatigue study showing conversion of **1c** (1 mM) to **1c2f** upon consecutive additions of **2f**. After each TCNE addition (20 equiv.), the system was allowed to relax for 48 h before the next addition.

CDs are flexible and are known to sometimes accommodate such bulky guests through partial or higher order-binding[60,61], in this particular case it still appears that the Diels–Alder adduct is not a good fit for the host. Metadynamics calculations under a sphere of confinement (GFN2-xTB level, see Supporting Information Section S9 for details), starting from a geometry where **1a2f** was forcibly positioned into the β-CD host cavity, led to fast ejection of the Diels–Alder adduct from the cavity (Supplementary Movie 1). A similar situation was observed with adduct **1a2g**, though longer residence time and partial binding was observed (Supplementary Movie 2, 3).

**Retro-Diels–Alder reaction studies**

Having established control over the host–guest dissociation, we speculated whether, under a different set of conditions, the inherent reversibility of Diels–Alder reactions could be leveraged to promote regeneration of the anthracene. Many previous reports of reversible pericyclic reactions use a

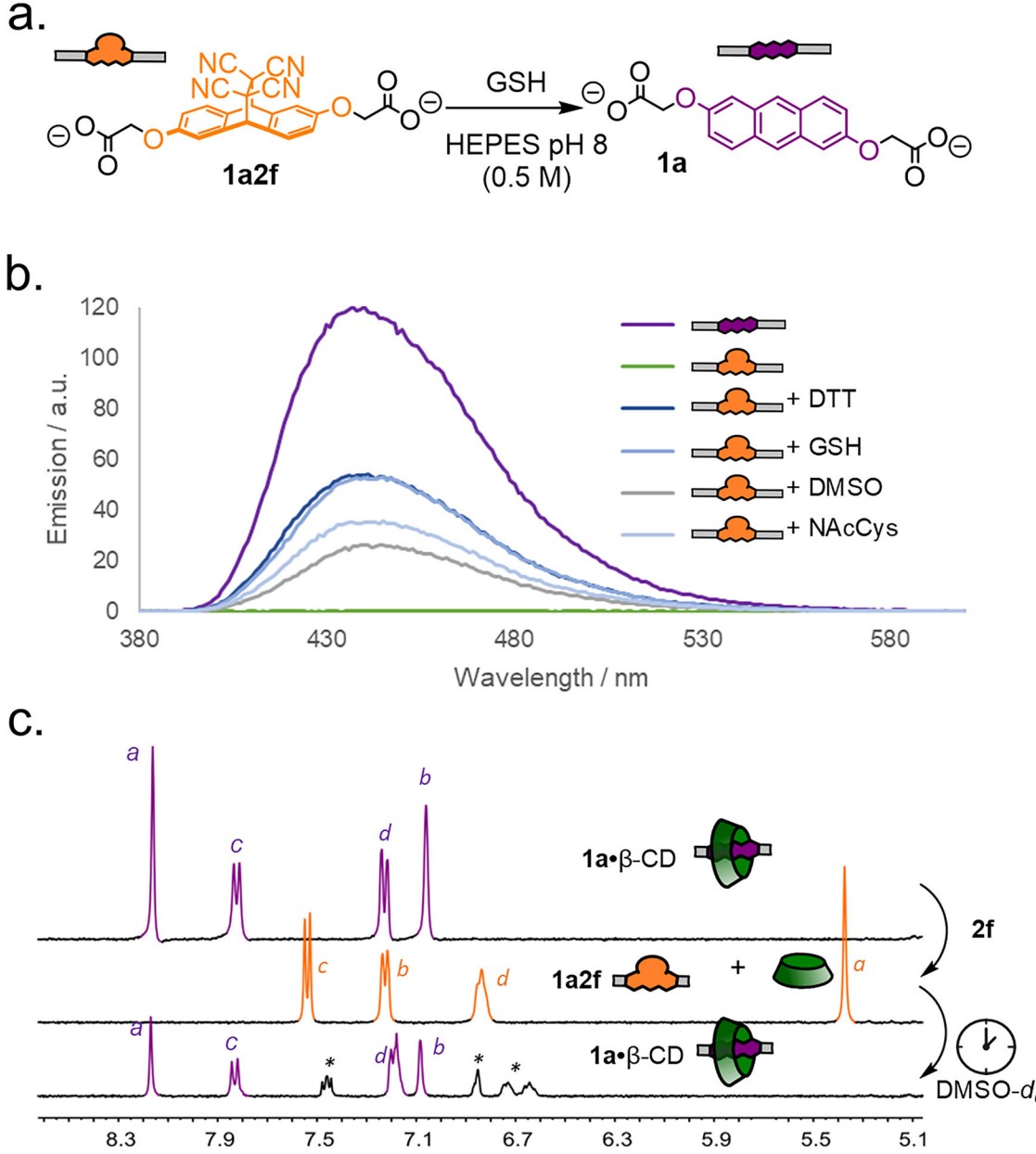

**Fig. 6 | Using selective alkene scavengers to switch between host–guest system states. a** Scheme of retro-Diels–Alder reaction of **1a2f** with GSH as alkene scavenger. **b** Fluorescence spectroscopy (100 μM, 0.5 M HEPES buffer, 298 K) data of **1a**, **1a2f** and **1a2f** + different nucleophiles (10 equiv.) following 2 h incubation. $\lambda_{ex}$ = 360 nm, Slit 2.5, 380–600 nm. **c** Truncated $^1$H-NMR spectra (400 MHz, D$_2$O/DMSO-$d_6$ 99:1, 298 K) of **1a**•β-CD (1 mM, top), the solution immediately after addition of TCNE **2f** (20 equiv., middle) and again after 48 h (bottom, asterix indicates unknown side product). See SI for further details.

rise in temperature to perturb the equilibrium position from the adduct towards the diene-dienophile pair[48–51]. As the broader goals of our research group are aimed at biological applications, we instead wanted to modulate the equilibrium position by addition of biologically relevant chemical stimuli. To achieve this, we decided to use an intercepting reaction (following Le Chatelier's principle) to perturb the product distribution towards the anthracene reactant. While Diels–Alder equilibria typically favour product[32,33], there is always free alkene/dienophile available in the equilibrated system. By adding a slower-reacting alkene scavenger[32,33,62], we aimed to modulate the equilibrium by gradually consuming the free dienophile, whilst simultaneously regenerating the anthracene.

An initial reactivity screen was conducted by simply mixing TCNE **2f** with different alkene scavengers in CDCl$_3$ (see Supplementary Table S2; solvent chosen to maximise solubility for all components). We found that

several nucleophiles could reliably achieve the intended alkene scavenging with complete conversion, including water-soluble thiols such as dithiothreitol (DTT), *N*-acetyl cysteine (NAcCys) and glutathione (GSH).

When trying to perform the same alkene scavenging starting from our previously generated Diels–Alder adducts, we came across an intriguing discovery. In an attempt to solubilise the molecule, we tried mixed the poorly water-soluble anthracene **1c** into water/DMSO mixtures. After adding TCNE **2f** (20 equiv.) to generate the TCNE-Diels–Alder adduct **1c2f**, we noticed that the solvent itself acted as a selective scavenger for TCNE **2f**. DMSO and TCNE are known to form a charge transfer pair, which here seems to initiate destructive radical polymerisation that consumes the free alkene[63]. This scavenging reaction is much slower than the pericyclic reaction between TCNE and anthracenes, and so a non-equilibrium steady-state is established. The system remains in the Diels–Alder adduct state while free

TCNE **2f** is available, but reverts to the free anthracene once all TCNE **2f** is consumed (Fig. 5a)[39].

We explored this phenomenon by adding 20 equivalents TCNE **2f** to neutral anthracene **1c** in pure DMSO-$d_6$. Full conversion to **1c2f** was observed within one minute, followed by gradual decay back to the initial anthracene **1c** over 72 h with full regeneration observed by [1]H-NMR spectroscopy (Fig. 5b). We cycled the scavenging reaction five times with only low fatigue observed (Fig. 5c).

To the best of our knowledge, this is the first discovery of a non-equilibrium steady state based on pericyclic reactions[64]. Given the importance of orthogonal reaction mechanisms in design of such non-equilibrium reaction networks, we believe this finding could be of interest to researchers in the field.

However, CD binding to hydrophobic guests is very weak in organic solvents like DMSO. To align better with our long-term goals, we hence investigated whether the scavenger-assisted retro-Diels–Alder reaction was compatible with aqueous media (Fig. 6a). We initially chose GSH as the nucleophile due to its biological importance, as upregulation of GSH levels are seen in conditions such as cancer, liver diseases and neurodegenerative disorders[65]. Retro-Diels–Alder reactions of both adducts **1a2f** and **1a2g** with GSH were carried out in HEPES (4-(2-hydroxyethyl)-1-piperazineethanesulfonic acid) buffer (0.5 M) to neutralise any acid buildup from the nucleophile scavenging step. In all cases, the retro-Diels–Alder reaction worked quickly (typically complete in 2 h for TCNE adducts), but with lower selectivity than in organic solvents. Between 30–60% of the anthracene could be recovered using this strategy as determined by [1]H-NMR spectroscopy and fluorescence measurements (Fig. 6b, Supplementary Figs. S29–31). DTT and NAcCys also initiated the retro-Diels–Alder reaction, as did running the reaction with DMSO-$d_6$ as a reactant. Using DMSO-$d_6$ as a co-solvent (1–8 vol%) also brought about the retro-Diels–Alder reaction in comparable efficiencies. Finally, we tried to conduct the retro-Diels–Alder reaction in the presence of β-CD to complete the cycle and regenerate the initial host–guest complex (Fig. 6c). Using a 99:1 D$_2$O/DMSO-$d_6$ mixture, we could regenerate the initial anthracene (and its host–guest complex) with the same efficiency (ca 50% yield) as without the added β-CD, validating the core hypothesis. The reason for the poor performance of the retro-Diels–Alder reaction in water and the nature of the formed side products are currently under investigation in our laboratory.

## Conclusions

In summary, we have demonstrated how a dynamic pericyclic reaction can be used to regulate cyclodextrin host–guest binding. The [4 + 2] cycloaddition reaction between dienophiles and 2,6-disubstituted anthracenes proceeds cleanly in water at ambient temperature, and gives full conversion to the Diels–Alder adduct within minutes (for TCNE **2f**) or hours (for maleimide **2g**). While the unmodified anthracenes are good binders for β-cyclodextrin, their corresponding Diels–Alder adducts were non-binding due to the increased steric bulk.

Finally, we disclosed a strategy for biasing the pericyclic reaction equilibrium towards the anthracene reactant. We used alkene scavengers to manipulate the equilibrium position, and by tuning the kinetics of the cycloaddition and scavenging reactions we could also achieve non-equilibrium steady-state behaviour.

These results hold potential for the construction of stimuli-responsive biomaterials and smart drug delivery systems. The mild aqueous conditions for the switching constitute a promising starting point for future biological applications. Given the strong links between this type of host–guest complexes and artificial molecular machines, we further believe these architectures to be valuable models for molecular machinery that can operate in a biorthogonal manner[66–68].

## Methods

### Representative protocol for forward Diels–Alder reaction
In an NMR tube with deuterated solvent (1 mL, D$_2$O or CDCl$_3$), anthracene was added (1 mM final concentration). Dienophile (10 equiv.) was added and the tube was left at the given reaction temperature in the dark under ambient atmosphere. The reaction was monitored over time using [1]H NMR spectroscopy and conversion was calculated by observing the ratio between product protons and the starting anthracene (most often central 9,10-anthracene protons at ~8.3 ppm). In all cases, clean conversion was observed. The NMR tube with the reaction was kept at the temperature given in Supplementary Table S1 for the full duration of the experiment.

### Retro-Diels–Alder reaction in DMSO-$d_6$
Anthracene was dissolved in DMSO-$d_6$ (1 mM) at room temperature under air in an NMR tube. TCNE (10 equiv.) was added in one batch and the tube was shaken once. The forward reaction was confirmed to be complete by the time a first [1]H-NMR spectrum could be recorded (<5 min). Retro reaction occurred spontaneously over 48–72 h, and conversion was monitored with NMR spectroscopy.

### Retro-Diels–Alder reaction in water
Reaction was initiated by addition of dienophile scavenger (10 equiv.) to a solution of the Diels–Alder adduct in D$_2$O (1 mM). The reaction proceeded spontaneously at ambient temperature and atmosphere without stirring. For fluorescence measurements, a similar protocol was followed, but an aliquot was taken from the reaction mixture and diluted with H$_2$O to 100 µM before measurement.

### Host–guest titrations
To a solution of anthracene **1a** (2 mM) in D$_2$O (with 8 mM NaOD to ensure deprotonation) in an NMR tube was added consecutive aliquots of cyclodextrin (10 mM) solution, and in between each addition [1]H-NMR spectra were recorded. The data was fitted using the anthracene protons at 8.3, 8.0, 7.3 and 4.6 ppm with a 1:1 binding model with bindfit (http://supramolecular.org).

## Data availability
Experimental procedures, optimisation data, NMR, fluorescence and mass spectra can be found in the electronic supporting information. Molecular dynamics raw data can be found in Supplementary Data 1, raw data for plots can be found in Supplementary Data 2 and NMR spectra of all new compounds in Supplementary Data 3.

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

## Acknowledgements

F.S. gratefully acknowledges financial support from University of Warwick, the Swedish Research Council (grant number 2020-04225), Stiftelsen Olle Engkvist Byggmästare (215-0407), Carl Tryggers stiftelse (21:1584), Magnus Bergvalls stiftelse and the KTH Royal Institute of Technology. D.R.S.P thanks the European Union for an individual Marie Skłodowska-Curie Actions Fellowship (No. 101150513), and the resources provided by the National Academic Infra-structure for Supercomputing in Sweden (NAISS) at the Tetralith cluster (NAISS 2024/22–342; NAISS 2025/22–653) for enabling the computations.

## Author contributions

The research project was conceived by F.S. and M.G. Synthesis work was led by M.G. and supported by A.R. Kinetic measurements and switching experiments were performed by M.G., A.R. and D.R.S.P. Theoretical calculations were performed by D.R.S.P. Data analysis was performed by all authors. The paper was written by F.S. with support from all authors.

## Funding

## Competing interests

F.S. is an Editorial Board Member for *Communications Chemistry*, but was not involved in the editorial review of, or the decision to publish this article.
