## [Transparent Peer Review file · Communications Chemistry]

Controlling cyclodextrin host-guest complexation in water with dynamic pericyclic chemistry

Corresponding Author: Dr Fredrik Schauffelberger

Version 0:

Reviewer comments:

Reviewer #1

(Remarks to the Author)

The manuscript deals with cyclodextrin inclusion reactions involving substituted anthracene derivatives in aqueous solutions, Diels-Alder cycloaddition reactions between 2,6-dioxyanthracenes and activated alkenes and alkynes, such reactions involving anthracene derivatives included in the cyclodextrin cavity, and retro Diels-Alder reaction studies. The topic of the paper is quite relevant and actual due to the reemergence of interest in cyclodextrin chemistry as a consequence of their numerous and increasing applications in pharmaceutical, medicinal, and food sciences. The paper is satisfactorily written, the results obtained seem sound and are in most cases properly discussed. However, some important information is missing, and numerous points need to be considered by the authors:

- Along with the ^1H NMR spectra recorded in the course of titrations, the corresponding binding isotherms and fitted lines should be presented so that reader can at least roughly assess the goodness of fit. Any information about the reproducibility of the experimental results?
- In the case of gamma-CD binding with compound 1a, the formation of the complex of 2:1 stoichiometry has been proposed with anthracene dimerization within the cavity. This finding should be discussed in more detail. In addition, the possibility of dimerization of anthracene derivatives in aqueous solutions should be considered, as this process introduces an additional equilibrium that must be accounted for in both the qualitative and quantitative description of all studied systems.
- Why was the temperature raised to observe complexation of beta-CD and compound 1d? That would be reasonable if the reaction were endothermic. However, that is usually not the case with the inclusion reactions of beta-CDs.
- $\text{pH} > 8$ is too rough; at least concentration of NaOD should be given.
- How was the 1a.beta-CD complex prepared? How do the authors know that in solution only the complex was present? What does it mean "an excess of TCNE 2f"? The amount of 2f added should be given. The experimental details about these experiments (including 1b) are missing.
- Generally, the compounds' concentrations should be given in all figure captions (e.g. c(1a) in the caption of Figure 3). The units of T and kobs are missing in Figures S1 and S3, respectively.
- The complete analysis of the kinetic data (including calculated kinetic traces) for all reactions should be presented in the SI. The intercept of the straight line in Figure S3, i.e. the rate constant of reverse reaction, should be commented.
- It should be stated that activation Gibbs energy of 23.2 kcal/mol corresponds to 20 °C.
- Was the inertness of HEPES buffer checked? Its concentration should be given.
- The values of excitation and emission slits, as well as that of excitation wavelength, are not specified for spectrofluorimetric experiments.
- It is not correct to generally conclude that "The [4+2] cycloaddition reaction between dienophiles and 2,6-disubstituted anthracenes in water is fast even at ambient temperature, ..."

Reviewer #2

(Remarks to the Author)

The authors present an exciting piece of work on employing a Diels-Alder (DA) pericyclic reaction to regulate the stability of supramolecular complexes between beta-CD and anthracene-derived guests. In a well-written and readable manuscript, they describe the synthesis of model guests, their binding properties with both natural and modified cyclodextrins, and [4+2] cycloadditions of several dienophiles with model anthracene guests, including kinetic studies of these reactions. Finally, the authors demonstrate the loss of affinity of the 1a2f adduct for beta-CD, the dynamic nature of the DE/retroDE

equilibrium under ambient conditions using several diene scavengers (without b-CD), and the operation of the complete system, where binding of the anthracene derivative to b-CD was restored upon addition of DMSO as a scavenger.

I believe the results are of interest to the supramolecular chemistry community and are suitable for publication in Communications Chemistry. However, I have several comments regarding the current version of the manuscript:

1) The most important point of criticism concerns the final part of the manuscript. After reading, it was not at all clear whether the authors had actually demonstrated the reversibility of the system, i.e., the loss of affinity due to the DE reaction and its subsequent restoration by using a diene scavenger in a cyclodextrin-containing mixture. The only sentence addressing this (line 256) can be easily overlooked. The main attention is drawn to Figure 6 and the accompanying text, which, however, only shows recovery of the anthracene derivative using a diene scavenger, but not in the presence of b-CD. I therefore strongly recommend replacing Figure 6 with Figure S19, which is highly important and directly demonstrates the key goal outlined in the manuscript title. In addition, rewriting the final section of the Retro-DA reaction studies chapter to emphasise this crucial experiment would greatly improve clarity and appeal of the text.

2) Since the anthracene derivatives used in this study demonstrate only moderate affinity towards b-CD, one can reasonably expect a significant portion of free guest and host in a 1:1 mixture at millimolar concentrations, as clearly visible in Figure 2. Therefore, it is essential to explicitly state the concentrations and H/G ratios used in each case. Please include this information both in the main text (e.g., line 182) and in the figure captions (e.g., Figure S18). Also, I recommend being more cautious when referring to these as “strong complexes” (line 92). Considering that b-CD binding affinities typically range between $\log K=0-6$, the complexes described in this study would be better classified as moderate affinity.

3) Please add peak labels to the top spectrum in Figure S19. In particular, clarify the identity of the peaks around 6.75 ppm and 7.60 ppm.

4) Please, consider expanding the discussion on the mechanism by which the DA reaction reduces the affinity of anthracene-derived guests. Does the DA reaction take place while the guest is bound inside the b-CD cavity, producing a non-binding product that is expelled? Or does the reaction occur only for unbound guests? Related to this, I am curious about the reaction rate of the DA process in the presence of b-CD. A proposed mechanistic scheme could be shown in Figure 1a.

5) I am not as surprised as the authors that 1d does not form an inclusion complex. The bulky tris-alkoxybenzene ring could serve as a stopper for b-CD in rotaxane structures. Please provide more details (in SI) on the computational analysis supporting the threading of the 1d terminus through the b-CD cavity.

Minor comments:

6) Line 148: There appears to be a typo. TCNE is not 2g.

7) Consider replacing the phrase “Complex binding” in Table 1, entry 3 with less ambiguous term such as “Intricate binding” or “Non-trivial binding”.

8) Figure 2: Consider using labels a, b, c,... instead of Ha, Hb, Hc,... Since it is already stated in the caption that these are ^1H NMR spectra, the “H” prefix is redundant.

9) Figure 4: consider adding H-atoms labels and signal assignment to improve clarity and readability.

10) Please consider adding NMR spectra of the dienophiles used in Figure S10 and onwards.

11) Figure S18: Please clarify the experimental setup. While the caption helps, it is not immediately obvious whether the top spectrum shows a 1b+b-CD mixture after addition of 2f or a 1b+2f mixture after the addition of b-CD. Using colour coding to differentiate the two independent experiments would be helpful.

Reviewer #3

(Remarks to the Author)

I quite like the idea of this manuscript, which is to report the use of reversible Diels-Alder cycloadditions as a mechanism to control reversible complexation in a beta-cyclodextrin host. Nonetheless, I have a few minor issues (and some more major ones) that combine to detract from my overall enthusiasm for this manuscript, and will need to be addressed before I can recommend publication.

Relatively minor comments include:

1. Abstract language – The phrase “primitive molecular switch” is vague. It is not clear what “primitive” is meant to connote. Similarly, the expression “regulating CD binding dynamically” is unclear. Both should be rephrased.

2. Claim of novelty – The authors state there are “no methods using selective chemical reactions to control binding.” This should be qualified. There are prior examples where host–guest binding in CDs has been regulated chemically (e.g. by pH, enzymatic, or covalent modification approaches).

3. Terminology – The manuscript sometimes uses “pericyclic reactions” and “cycloadditions” interchangeably, although the latter is a subclass of the former. Precision would improve clarity.

4. Figure caption – Figure 1 currently reads “cartoon on.” This should be corrected to “cartoon of” (or “schematic of”).

5. Retro reaction terminology – The retro-Diels–Alder process is referred to as “retro-reactions.” The proper term is “retro-Diels–Alder reaction.”

6. Alkene scavenger terminology – Nucleophiles are referred to as “alkene scavengers,” but the reaction mechanism and products are not characterized. A more neutral phrasing should be used, or supporting evidence for the products provided.

7. Water solubility explanation – The explanation of why certain guests did not bind due to solubility issues is not clear. CD inclusion itself often enhances solubility, and this point requires clarification.

More major concerns include:

1. Overstatement of host–guest size restrictions: The authors claim that Diels–Alder adducts cannot bind β -CD because their

calculated dimensions are close to the nominal cavity diameter, and that “guests generally need to be much smaller than the cavity size for complementary fit.” This conclusion is inaccurate and oversimplified. Cyclodextrin cavities are well known to be structurally flexible, capable of deformation and induced fit, and to bind guests considerably larger than their nominal cavity diameter. Examples include tamoxifen complexes with β -CD derivatives (Bilensoy et al., *J. Incl. Phenom. Macrocycl. Chem.*, 2007; Shukla et al., *J. Radioanal. Nucl. Chem.*, 2009; Buchanan et al., *J. Pharm. Sci.*, 2006) and other bulky drugs such as tricyclic antidepressants (Beilstein *J. Org. Chem.*, 2017). These cases demonstrate that partial inclusion, asymmetric deformation, and 2:1 host:guest stoichiometries are common. The sweeping claim should therefore be either removed or revised substantially with acknowledgment of this established body of work.

2. Binding constant reporting: Table 1 presents K_a values without error estimates or methodological details. From the ESI it is apparent that a 1:1 binding model and BindFit were used but this is not stated in the main text. A footnote should be added specifying the fitting method, model, and the associated uncertainties. Without this, the quantitative values are misleadingly precise.

3. There are several mechanism-focused overstatements scattered throughout the manuscript, including:

o The authors claim that agreement of their Eyring parameters with literature “indicates that the reaction proceeds through a similar pathway.” This is not supported by data; similar parameters do not establish mechanism.

o The description that “many Diels–Alder reactions in water are accelerated through a combination of hydrogen-bond activation and the hydrophobic effect” is an oversimplification of the nuanced effects of water on pericyclic chemistry.

o Conversion is said to be confirmed by “large NMR shifts.” While these shifts are consistent with conversion, they are not definitive evidence. HRMS data is provided in the SI and should be explicitly referenced in the main text to substantiate the claim.

4. Conceptual clarity and terminology: Several descriptors are unclear or unsupported, including:

o “Prototypical dienophile” (dimethyl fumarate) – please clarify the basis for this claim.

o “Typical biorthogonal reactivity profiles” – too vague, requires examples or a citation.

o “Pseudorotaxane structure” to describe a β -CD•anthracene complex – debatable usage; if retained, it should be justified with references.

5. Inconsistent solvent use: The manuscript reports host–guest titrations and kinetic experiments in D₂O, DMSO-*d*₆, and CDCl₃ without a consistent rationale. The choice of solvent and its impact on solubility, binding, and reaction outcomes should be explained.

In summary, the manuscript presents an interesting and potentially valuable approach to modulating CD host–guest binding via reversible Diels–Alder chemistry. However, the text currently suffers from imprecise language, oversimplified mechanistic interpretations, and overstatement of conclusions relative to the data presented. Most critically, the sweeping claims about cyclodextrin binding limitations (i.e., that guests must be “much smaller” than the cavity) are not supported by the broader literature and ignore the well-documented structural flexibility of CDs.

This manuscript will be significantly improved if the authors:

- Qualify their mechanistic claims,
- Provide methodological details and error estimates for K_a values,
- Soften or revise their conclusions about host–guest size restrictions, integrating the established literature on CD flexibility (including tamoxifen and bulky aromatic complexes),
- And improve clarity in language and terminology throughout.

At present, the work lacks sufficient experimental support for several of its most sweeping claims. The authors should either (i) provide additional experimental data to substantiate these claims, or (ii) significantly moderate their language to more accurately reflect the limitations of their evidence. I am in favor of the first approach, as the underlying idea is intriguing and deserving of rigorous validation.

Reviewer #4

(Remarks to the Author)

I co-reviewed this manuscript with one of the reviewers who provided the listed reports. This is part of the Communications Chemistry initiative to facilitate training in peer review and to provide appropriate recognition for Early Career Researchers who co-review manuscripts.

Version 1:

Reviewer comments:

Reviewer #1

(Remarks to the Author)

The authors have taken into account all the reviewers' comments and revised the manuscript accordingly. Therefore, it can be accepted in its present form.

Reviewer #2

(Remarks to the Author)

I thank the authors for considering my comments and revising the manuscript accordingly. I believe that the current version of the manuscript is significantly clearer and more readable, and I recommend its publication in Communications Chemistry.

Reviewer #3

(Remarks to the Author)

The authors have done an excellent job responding to the comments that I made on the previous version of this manuscript. My only remaining concern is relatively minor:

In response to my concern about the poor solubility of compound 1c, the authors responded that the use of β -cyclodextrin (β -CD) as a solubilizing supramolecular host did not improve the solubility of guest 1c. However, β -CD itself has relatively poor water solubility and is therefore not an ideal choice for evaluating whether cyclodextrin inclusion could enhance guest dissolution. Since the manuscript already includes data involving methyl- β -cyclodextrin and γ -cyclodextrin—both of which are substantially more soluble and frequently used for solubilization—it would be interesting (though not essential) to see whether either of those more soluble hosts enhances the solubility of 1c.

Reviewer #4

(Remarks to the Author)

I co-reviewed this manuscript with one of the reviewers who provided the listed reports. This is part of the Communications Chemistry initiative to facilitate training in peer review and to provide appropriate recognition for Early Career Researchers who co-review manuscripts.

Reviewer #1:

“The manuscript deals with cyclodextrin inclusion reactions involving substituted anthracene derivatives in aqueous solutions, Diels-Alder cycloaddition reactions between 2,6-dioxyanthracenes and activated alkenes and alkynes, such reactions involving anthracene derivatives included in the cyclodextrin cavity, and retro Diels-Alder reaction studies. The topic of the paper is quite relevant and actual due to the reemergence of interest in cyclodextrin chemistry as a consequence of their numerous and increasing applications in pharmaceutical, medicinal, and food sciences. The paper is satisfactorily written, the results obtained seem sound and are in most cases properly discussed.”

We thank the referee for the positive assessment of our work and the insightful and constructive comments.

“Along with the ¹H NMR spectra recorded in the course of titrations, the corresponding binding isotherms and fitted lines should be presented so that reader can at least roughly assess the goodness of fit. Any information about the reproducibility of the experimental results?”

We are grateful for the useful suggestions. Binding isotherms and fitted lines have all been added to the SI in the form of the bindfit links, added in association to each measurement as part of the caption. We have incorporated this using the general format: “The binding isotherms of goodness-of-fit for this experiment can be accessed at <LINK>.” Below follows also the links for the convenience of the referees:

1a + β-CD:

<http://app.supramolecular.org/bindfit/view/3700ef84-84be-46e5-bef1-3d090ca3886a>

1a + β-CD-OMe:

<http://app.supramolecular.org/bindfit/view/0ce101d6-5a6f-4869-b79a-f71c47d0a65e>

1a + β-CD-N₃:

<http://app.supramolecular.org/bindfit/view/298d553b-3c8e-46f2-be29-26253465df9d>

1b + β-beta CD:

<http://app.supramolecular.org/bindfit/view/37de052f-2c10-4177-8ea8-d8505cc33a2b>

As for reproducibility, to probe this we repeated the core experiment between **1a** and β-CD to perform a triplicate titration. The results yielded values of 3741.8 M⁻¹, 3587.3 M⁻¹ and 3498.7 M⁻¹, or a mean of (3.6 ± 0.79) × 10³ M⁻¹. The individual BindFit links for these experiments are as follows:

1st run: <http://app.supramolecular.org/bindfit/view/5e196953-4209-4a53-a3ae-26b5bc95edb0>

2nd run: <http://app.supramolecular.org/bindfit/view/74b1ef49-fb09-4c25-bb76-6a04cff46265>

3rd run: <http://app.supramolecular.org/bindfit/view/ce6d0f6f-73db-48e4-a8d5-cdd4a0765076>

“In the case of gamma-CD binding with compound 1a, the formation of the complex of 2:1

stoichiometry has been proposed with anthracene dimerization within the cavity. This finding should be discussed in more detail.”

We are currently investigation this equilibrium in-depth as part of a follow-up study to this work. Our newer findings corroborate the preliminary results reported herein, i.e. γ -CD gives rise to a mixture between 1:1 and 2:1 binding modes with anthracenes (one in fast and one in slow exchange), and inclusion significantly increases capacity for photodimerization via the [4+4] addition (as previously reported in supporting information reference 7 – *Tetrahedron*, **1987**, 43, 1485–1494). As the scope of this initial report is already quite wide, we believe a detailed discussion of these issues is better suited to this follow up work. We hope the referee understand this and are of course happy to share further preliminary data should the referee request so.

“In addition, the possibility of dimerization of anthracene derivatives in aqueous solutions should be considered, as this process introduces an additional equilibrium that must be accounted for in both the qualitative and quantitative description of all studied systems.”

No anthracene dimerization was observed in any of the β -CD-based experiments presented in the manuscript. All solutions were prepared fresh and kept in the dark during use. The [4+4] addition product produces some distinct signals in an otherwise silent NMR region (ca 6-6.5 ppm), so the absence of the photodimer is simple to verify. The key decomposition pathway observed upon leaving the solution standing in the light is actually photooxidation rather than dimerisation, which is another typical side reaction that occurs with anthracenes.

“Why was the temperature raised to observe complexation of beta-CD and compound 1d? That would be reasonable if the reaction were endothermic. However, that is usually not the case with the inclusion reactions of beta-CDs.”

Heating the reaction mixture is common practice in the field of mechanically interlocked molecules, when using macrocycles with cavity sizes close to the stoppering size of the guest extremities. The rise in temperature is not intended to change the binding strength (though this parameter will also be influenced) but to accelerate the rate of complexation. We have clarified that this in the main text.

“pH > 8 is too rough; at least concentration of NaOD should be given.”

We used [NaOD] = 8 mM throughout. We have replaced all instances where “pH > 8” was mentioned with this value.

“How was the 1a.beta-CD complex prepared? How do the authors know that in solution only the complex was present? What does it mean “an excess of TCNE 2f”? The amount of 2f added should be given. The experimental details about these experiments (including 1b) are missing.”

We have increased the level of detail in the main text regarding the preparation of the host-guest complex. The preparation information is also present in the SI under the headline “Host-Guest titrations”:

“To a solution of anthracene **1a** (2 mM) in D₂O (with 8 mM NaOD to ensure deprotonation) in an NMR tube was added consecutive aliquots of cyclodextrin (10 mM) solution, and in between each addition ¹H-NMR spectra were recorded. The data was fitted using the anthracene protons at 8.3, 8.0, 7.3 and 4.6 ppm with a 1:1 binding model with bindfit (<http://supramolecular.org>).^{[5,6]”}

We are fairly confident that only the 1:1 complex and the uncomplexed host/guest pair are present in the system. This is because the system is clearly in fast exchange as per NMR measurements with no other new peaks appearing. Mapping the exchange process gives a good fit to a 1:1 model (see BindFit links above), and the cavity size is not large enough to accommodate higher-order binding modes. This is in line with observations from previous literature, for example the review in reference 12 (new numbering) and references therein.

We have changed all instances where the expression “excess TCNE” was used to “TCNE (20 equiv.)”, which is the actual value added.

“Generally, the compounds’ concentrations should be given in all figure captions (e.g. c(1a) in the caption of Figure 3). The units of T and k_{obs} are missing in Figures S1 and S3, respectively.”

We thank the referee for pointing out this omission. The concentrations in questions are:

c(**1a**) in figure 3 is 2mM

c(**1a**) in figure 4 is 1mM

c(**1c**) in figure 5 is 1mM

c(**1a**) in figure 6 is 1mM

These values are now added to the figures. We have also added units of T (Kelvin) and k_{obs} (h⁻¹) to the axes in Figures S1 and S3.

“The complete analysis of the kinetic data (including calculated kinetic traces) for all reactions should be presented in the SI. The intercept of the straight line in Figure S3, i.e. the rate constant of reverse reaction, should be commented.”

We have added the requested kinetic data to the SI as new figures S4-S7. To ensure conformity within the ESI, we have relabelled Section S5.2 as “Kinetic data” and added this information under a new subheading. Note that for the reaction of **1a** and **2e**, the mixture was heterogenous and so we did not record a kinetic trace, as this would not accurately represent the reaction profile.

The following figures and captions have been added to the SI as new Figures S4-S7:

Figure S4. Initial rate data for the reaction of anthracene **1a** (1 mM) with activated alkene **2a** (20 mM) in D₂O (with 8 mM NaOD).

Figure S5. Plot of rate data using pseudo-first order reaction model for reaction between anthracene **1a** (1 mM) and activated alkene **2a** (20 mM) as per Figure S4.

Figure S6. Initial rate data for the reaction of anthracene **1a** (1 mM) with maleimide **2g** (8.1 mM) in D₂O (with 8 mM NaOD).

Figure S7. Plot of rate data using pseudo-first order reaction model for reaction between anthracene **1a** (1 mM) and maleimide **2g** (20 mM) as per Figure S6.

That the intercept of the straight line is nonzero demonstrates that the reaction is reversible, as is well-known for the Diels-Alder reaction (see for example <https://doi.org/10.1002/kin.20960>). We have added a note to this effect in the caption for Figure S3.

“It should be stated that activation Gibbs energy of 23.2 kcal/mol corresponds to 20 °C.”

The requested clarification has been added.

“Was the inertness of HEPES buffer checked? Its concentration should be given.”

500 mM HEPES buffer was used, and a note about this has been added. We consistently observed less side reactions and higher recovery of the initial fluorescence in the HEPES buffer than in non-buffered solution, indicating inertness.

“The values of excitation and emission slits, as well as that of excitation wavelength, are not specified for spectrofluorimetric experiments.”

This info was previously available in the **S2 General experimental** section. On the request of the referee, we have now also added the values to the caption of Figure 6.

“It is not correct to generally conclude that “The [4+2] cycloaddition reaction between dienophiles and 2,6-disubstituted anthracenes in water is fast even at ambient temperature, ...”
“

We apologise for the sweeping nature of this statement. We have changed this sentence to:

“The [4+2] cycloaddition reaction between dienophiles and 2,6-disubstituted anthracenes proceeds cleanly in water at ambient temperature...”

Reviewer #2 (Remarks to the Author):

“The authors present an exciting piece of work on employing a Diels-Alder (DA) pericyclic reaction to regulate the stability of supramolecular complexes between beta-CD and anthracene-derived guests. In a well-written and readable manuscript, they describe the synthesis of model guests, their binding properties with both natural and modified cyclodextrins, and [4+2] cycloadditions of several dienophiles with model anthracene guests, including kinetic studies of these reactions.

Finally, the authors demonstrate the loss of affinity of the 1a2f adduct for b-CD, the dynamic nature of the DE/retroDE equilibrium under ambient conditions using several diene scavengers (without b-CD), and the operation of the complete system, where binding of the anthracene derivative to b-CD was restored upon addition of DMSO as a scavenger.

I believe the results are of interest to the supramolecular chemistry community and are suitable for publication in Communications Chemistry.”

We thank the referee for this positive assessment and their insightful and constructive comments.

“1) The most important point of criticism concerns the final part of the manuscript. After reading, it was not at all clear whether the authors had actually demonstrated the reversibility of the system, i.e., the loss of affinity due to the DE reaction and its subsequent restoration by using a diene scavenger in a cyclodextrin-containing mixture. The only sentence addressing this (line 256) can be easily overlooked. The main attention is drawn to Figure 6 and the accompanying text, which, however, only shows recovery of the anthracene derivative using a diene scavenger, but not in the presence of β -CD. I therefore strongly recommend replacing Figure 6 with Figure S19, which is highly important and directly demonstrates the key goal outlined in the manuscript title. In addition, rewriting the final section of the Retro-DA reaction studies chapter to emphasise this crucial experiment would greatly improve clarity and appeal of the text.”

We agree with the referee that this section was not clear enough in the previous iteration of the manuscript. We did indeed conduct preliminary experiments of several retro-Diels-Alder reactions in the presence of β -CD as well. However, the interpretation of these experiments is made difficult by the apparent side reactions occurring during the retro-Diels-Alder reaction in water. The unidentified side products (*vide infra*) also appeared to exhibit some degree of CD binding, which complicates analysis. The experiment shown in Figure S26 (new numbering) does clearly show recovery of the original, complexed anthracene in the presence of β -CD, but with incomplete recovery. Still, we see the point of the referee and agree that the experiment does represent a logical conclusion to the narrative – even if the system needs further development to reach its true potential. We have hence reworded the discussion in this part as suggested by the referee, and also moved part of Figure S26 to the main manuscript as new Figure 6c (previous Figure 6c moved to Fig 6b, previous Fig 6b removed):

*“Finally, we tried to conduct the retro-Diels-Alder reaction in the presence of β -CD to complete the cycle and regenerate the initial host-guest complex. Using a 99:1 D₂O/DMSO-*d*₆ mixture, we could regenerate the initial anthracene (and its host-guest complex) with the same efficiency (ca 50% yield) as without the added β -CD, validating the core hypothesis (Figure 6c). The reason for the poor performance of the retro-Diels-Alder reaction in water and the nature of the formed side products are currently under investigation in our laboratory.”*

“2) Since the anthracene derivatives used in this study demonstrate only moderate affinity towards β -CD, one can reasonably expect a significant portion of free guest and host in a 1:1 mixture at millimolar concentrations, as clearly visible in Figure 2. Therefore, it is essential to explicitly state the concentrations and H/G ratios used in each case. Please include this information both in the main text (e.g., line 182) and in the figure captions (e.g., Figure S18).”

The requested changes have been made and we have included the concentrations of all relevant species throughout the text.

“I recommend being more cautious when referring to these as “strong complexes” (line 92). Considering that b-CD binding affinities typically range between $\log K=0-6$, the complexes described in this study would be better classified as moderate affinity.”

We see the point of the referee, and have replaced the wording “strong complexes” with weaker alternatives throughout.

“3) Please add peak labels to the top spectrum in Figure S19. In particular, clarify the identity of the peaks around 6.75 ppm and 7.60 ppm.”

Peaks labels have been added to Figure S26 (new numbering) to the extent possible. Please note that the identify of the side product generated in this experiment and also in other retro-Diels-Alder experiments done in water (see Fig 6c) is currently unknown, despite extensive efforts on our part to identify the generated side product *in situ* via NMR spectrometry, mass spectrometry and HPLC (and also to isolate the species). The dynamic nature of the system as a whole significantly complicates all such analytical efforts. The identity of the peaks around 6.75 ppm and 7.60 ppm is hence unknown. We have added a note about this in the figure caption.

“4) Please, consider expanding the discussion on the mechanism by which the DA reaction reduces the affinity of anthracene-derived guests. Does the DA reaction take place while the guest is bound inside the b-CD cavity, producing a non-binding product that is expelled? Or does the reaction occur only for unbound guests? Related to this, I am curious about the reaction rate of the DA process in the presence of b-CD. A proposed mechanistic scheme could be shown in Figure 1a.”

We have expanded the discussion around the mechanism of the CD affinity reduction via DA reaction, in line with this comment and the similar comment from referee 3. To gain further insights in this area, we performed the suggested measurements and measured the rates of the Diels-Alder reaction between **1a** with both **2f** and **2g** in the presence of an excess of β -CD. We did not observe any significant decrease in rate for either reaction, with the fast reaction with **2f** still happening faster than NMR timescale can measure and the slower reaction with **2g** producing next to identical kinetic trace with and without CD (see below). This indicates that the reaction happen in or close to the cyclodextrin cavity (as there is no concentration-dependant rate decrease), but without any synergistic effects stemming from the CD binding. We have added these results to the main manuscript and discuss them qualitatively.

Supporting Figure 1. Kinetic profiles for the reaction of **1a** (2 mM) and **2g** (20 mM) in D₂O (with 8 mM NaOD) at RT, without and with β -CD (22 equiv.)

“5) I am not as surprised as the authors that **1d** does not form an inclusion complex. The bulky tris-alkoxybenzene ring could serve as a stopper for β -CD in rotaxane structures. Please provide more details (in SI) on the computational analysis supporting the threading of the **1d** terminus through the β -CD cavity.”

We investigated this aspect computationally via the typical computational workflow we use for rotaxane synthesis, when we investigate whether the stoppering power is large enough to create a true mechanical bond. This works by crudely drawing the host-guest complex in a (pseudo)rotaxane state, then conducting initial geometry optimisation via low-level methods. Afterwards, we run molecular metadynamics along the workflow described in the computational section (GFN2-xTB level of theory) to identify either direct slippage (sometimes hard to observe even with metadynamics if binding is favorable) or slippage-prone conformations. For this example, we observed several conformations where it is clear that the lumen of the flexible cyclodextrin moiety has clearly eclipsed the full width of the stopper, see for example the image below:

Supporting Figure 2. Snapshot of metadynamic simulation showing a slippage-prone conformation of **1d**• β -CD.

We have found predictions made via this workflow generally translates to wet chemistry, and hence we did expect **1d** to be able to bind β -CD. However, there are likely kinetic barriers to complexation, as there are only very limited conformations where slippage or threading can occur. We have added a reference to the SI section S9 in the main text regarding this, along with a note on our workflow into the computational section S9 in the SI.

“6) Line 148: There appears to be a typo. TCNE is not 2g.”

Corrected, thanks for pointing out this error.

“7) Consider replacing the phrase “Complex binding” in Table 1, entry 3 with less ambiguous term such as “Intricate binding” or “Non-trivial binding”.”

We have replaced the term “complex” binding with “non-trivial binding” as suggested by the referee.

8) Figure 2: Consider using labels a, b, c,... instead of Ha, Hb, Hc,... Since it is already stated in the caption that these are ^1H NMR spectra, the “H” prefix is redundant.

We have changed the labels to “a,b,c...” instead of “Ha,Hb,Hc...” in line with the referee’s suggestions.

“9) Figure 4: consider adding H-atoms labels and signal assignment to improve clarity and readability.”

We have added signal assignment on the molecular structure, and have changed the existing H-atom labels according to the request in point 8 to make them stand out more.

“10) Please consider adding NMR spectra of the dienophiles used in Figure S10 and onwards.”

We have added the requisite ^1H -NMR spectra of TCNE **2f** and maleimide **2g** as new Figures S19-S20.

“11) Figure S18: Please clarify the experimental setup. While the caption helps, it is not immediately obvious whether the top spectrum shows a 1b+b-CD mixture after addition of 2f or a 1b+2f mixture after the addition of b-CD. Using colour coding to differentiate the two independent experiments would be helpful.”

We apologise for the confusion, and agree the clarity of this experiment (new figure S25) needed to be improved. We have followed the referee’s suggestion and colour coded the respective two experiments, added sections, cartoons and additional arrows, and added additional detail in the caption to explain what it is we have measured.

Reviewer #3 (Remarks to the Author):

“I quite like the idea of this manuscript, which is to report the use of reversible Diels-Alder cycloadditions as a mechanism to control reversible complexation in a beta-cyclodextrin host. Nonetheless, I have a few minor issues (and some more major ones) that combine to detract from my overall enthusiasm for this manuscript, and will need to be addressed before I can recommend publication.”

We thank the referee for the positive assessment of the manuscript, and are very grateful for the formative feedback. We have tried to address all independent aspects of the issues spotted by the referee.

“1. Abstract language – The phrase “primitive molecular switch” is vague. It is not clear what “primitive” is meant to connote. Similarly, the expression “regulating CD binding dynamically” is unclear. Both should be rephrased.”

We have rephrased as indicated and acknowledge the vagueness of the previous terms. We have removed “primitive” from the first term, and have changed “regulating CD binding dynamically” to “controlling CD binding”.

“2. Claim of novelty – The authors state there are “no methods using selective chemical reactions to control binding.” This should be qualified. There are prior examples where host–guest binding in CDs has been regulated chemically (e.g. by pH, enzymatic, or covalent modification approaches).”

There are indeed methods using selective enzyme cleavage of a guest that can be used to control binding, and upon further elaboration we agree this should qualify as a “selective chemical reaction for controlled binding”. We thank the referee for pointing out the inaccuracy of this initial statement and have rephrased this sentence to say that “there are few methods for controlled binding using selective chemical reactions” (followed by new references 6a and 6b). We apologise for the initial inaccurate statement and hope the referee agrees that the new phrasing places the work in a better context novelty-wise.

The new references added are:

6. a) Wankar, J., Kotla, N. G., Gera, S., Rasala, S., Pandit, A. & Rochev, Y. A. Recent Advances in Host–Guest Self-Assembled Cyclodextrin Carriers: Implications for Responsive Drug Delivery and Biomedical Engineering. *Adv. Funct. Mater.* **30**, 1909049 (2020); b) Gil, E. S. & Hudson, S. M. Stimuli-responsive polymers and their bioconjugates. *Prog. Polym. Sci.* **29**, 1173–1222 (2004).

“3. Terminology – The manuscript sometimes uses “pericyclic reactions” and “cycloadditions” interchangeably, although the latter is a subclass of the former. Precision would improve clarity.”

We thank the referee for pointing this out. We have changed most instances of “cycloadditions” to “pericyclic reactions” for consistency, except when specifically discussing the [4+2] Diels-Alder reaction where using the specified term “cycloaddition” is more appropriate.

“4. Figure caption – Figure 1 currently reads “cartoon on.” This should be corrected to “cartoon of” (or “schematic of”).”

Corrected.

“5. Retro reaction terminology – The retro-Diels–Alder process is referred to as “retro-reactions.” The proper term is “retro-Diels–Alder reaction.””

Corrected.

“6. Alkene scavenger terminology – Nucleophiles are referred to as “alkene scavengers,” but the reaction mechanism and products are not characterized. A more neutral phrasing should be used, or supporting evidence for the products provided.”

As mentioned in the supporting information section S7.5, we did observe the Michael adduct/alkene scavenger product from **2f** and GSH by mass spectrometry. Given the wide literature precedence for Michael addition of thiols to activated alkenes such as those used herein, we believe this term is appropriately used, as its descriptive nature helps the reader better understand the system operation.

“7. Water solubility explanation – The explanation of why certain guests did not bind due to solubility issues is not clear. CD inclusion itself often enhances solubility, and this point requires clarification.”

We did try to check the solubility of **1c** also after β -CD addition, and the system was still not soluble at the same concentrations as otherwise used in Table 1. This was the only guest that did not bind due to solubility issues, as the other guests were fully water-soluble. We have added a note stating that β -CD did not significantly improve the water-solubility of **1c**.

*“1. Overstatement of host–guest size restrictions: The authors claim that Diels–Alder adducts cannot bind β -CD because their calculated dimensions are close to the nominal cavity diameter, and that “guests generally need to be much smaller than the cavity size for complementary fit.” This conclusion is inaccurate and oversimplified. Cyclodextrin cavities are well known to be structurally flexible, capable of deformation and induced fit, and to bind guests considerably larger than their nominal cavity diameter. Examples include tamoxifen complexes with β -CD derivatives (Bilensoy et al., *J. Incl. Phenom. Macrocycl. Chem.*, 2007; Shukla et al., *J. Radioanal. Nucl. Chem.*, 2009; Buchanan et al., *J. Pharm. Sci.*, 2006) and other bulky drugs such as tricyclic antidepressants (Beilstein *J. Org. Chem.*, 2017). These cases demonstrate that partial inclusion, asymmetric deformation, and 2:1 host:guest stoichiometries are common. The sweeping claim*

should therefore be either removed or revised substantially with acknowledgment of this established body of work.”

This is a good point from the referee. In our initial discussion, we never meant to imply that asymmetric deformation, higher order binding and partial inclusion was not possible features of the equilibria observed. In this case, the NMR experiments described in Figure 4 clearly shows that the CD is at least much more weakly bound than before (i.e. limited spectral shift change for H3/5), but upon further consideration of the referee’s point we see the need for a revised text in this section. We have changed the order of the text in the associated section to begin with the NMR measurements and then talk about how the computations corroborate this data. We have substantially weakened the statement about host-guest size restrictions and removed the reference to the 55% rule, and instead added a sentence explaining how CD binding is flexible, referenced some of the suggested studies on binding of bulky guests from the referee and added that care must be taken in the interpretation of the results. We sincerely thank the referee for this useful suggestions, and hope they agree that the revised text avoids the oversimplification of the last version.

“2. Binding constant reporting: Table 1 presents K_a values without error estimates or methodological details. From the ESI it is apparent that a 1:1 binding model and BindFit were used but this is not stated in the main text. A footnote should be added specifying the fitting method, model, and the associated uncertainties. Without this, the quantitative values are misleadingly precise.”

See response to referee 1. We have added details to the footnote in Table 1, added error values and expanded the corresponding sections in the ESI, including BindFit links.

“o The authors claim that agreement of their Eyring parameters with literature “indicates that the reaction proceeds through a similar pathway.” This is not supported by data; similar parameters do not establish mechanism.”

This is true, we have revised the statement and removed the suggestion about pathway similarity.

“o The description that “many Diels–Alder reactions in water are accelerated through a combination of hydrogen-bond activation and the hydrophobic effect” is an oversimplification of the nuanced effects of water on pericyclic chemistry.”

We acknowledge the complex and still partially unknown nature of the rate accelerations of Diels-Alder reactions in water, and that a more nuanced explanation is needed. We have changed this sentence to read:

“Diels-Alder reactions in water often proceed faster than in organic solvents, and a number of explanations for this phenomena – such as hydrogen-bond activation and the hydrophobic effect – have been put forward.”

“o Conversion is said to be confirmed by “large NMR shifts.” While these shifts are consistent

with conversion, they are not definitive evidence. HRMS data is provided in the SI and should be explicitly referenced in the main text to substantiate the claim.”

We have added references to the complementary HRMS evidence in the main text, including reference to the corresponding supporting figures.

“4. Conceptual clarity and terminology: Several descriptors are unclear or unsupported, including:

- o “Prototypical dienophile” (dimethyl fumarate) – please clarify the basis for this claim.*
- o “Typical biorthogonal reactivity profiles” – too vague, requires examples or a citation.*
- o “Pseudorotaxane structure” to describe a β -CD•anthracene complex – debatable usage; if retained, it should be justified with references.”*

We have revised the unclear or unsupported terms, and rewritten the text to clarify our points in each case. We have left one reference to the pseudorotaxane structure when discussing the geometric orientation, as this term is well-adopted to describe the orientation where a guest is threaded through a macrocyclic host and the termini are protruding a significant distance out from each side of the macrocycle.

“5. Inconsistent solvent use: The manuscript reports host–guest titrations and kinetic experiments in D₂O, DMSO-d₆, and CDCl₃ without a consistent rationale. The choice of solvent and its impact on solubility, binding, and reaction outcomes should be explained.”

We apologise for not making the rationale behind the solvent usage clear enough in the previous version of the manuscript. We conducted studies for both forward and retro-Diels-Alder reactions in CDCl₃ as well as in D₂O to compare the reaction efficiency in both solvents. Obviously, there is no CD binding in halogenated organic solvents as the binding is driven to a large extent by the hydrophobic effect. We hence moved to aqueous experiments as soon as reactivity was confirmed in CDCl₃. We tried to explain that the use of DMSO came about when trying to solubilise PEG-containing anthracene **1c**, and once we discovered its capabilities as a scavenger we started using it as a reactant rather than solvent. A higher DMSO proportion than ca 10% in a water mixture will effectively destroy any binding interactions, which is why we kept the amount low. We have tried to clarify these points in the revised version of the manuscript.

“This manuscript will be significantly improved if the authors:

- Qualify their mechanistic claims,*
- Provide methodological details and error estimates for K_a values,*
- Soften or revise their conclusions about host–guest size restrictions, integrating the established literature on CD flexibility (including tamoxifen and bulky aromatic complexes),*
- And improve clarity in language and terminology throughout.”*

We thank the referee for their rigorous evaluation and have aimed to incorporate all their suggested changes in our revised manuscript. We believe the constructive criticism and great insights provided by the referee has significantly improved the manuscript, and are grateful for the superb feedback.

“At present, the work lacks sufficient experimental support for several of its most sweeping claims. The authors should either (i) provide additional experimental data to substantiate these claims, or (ii) significantly moderate their language to more accurately reflect the limitations of their evidence. I am in favor of the first approach, as the underlying idea is intriguing and deserving of rigorous validation.”

As the referee can see, we have largely tried to follow their first suggested approach by adding more experimental details, conducting additional mechanistic experiments and revising and expanding the discussion on the binding changes by size modulation. We note however that it is challenging to fully determine the mechanism of binding/lack of binding of the Diels-Alder adducts like **1a2f**, as binding is clearly weak. Here, we have instead tried to soften the language to acknowledge our lack of evidence for this point.

Unsolicited changes:

During editing of this revision, we spotted three errors in our original reporting that we have now corrected:

- In the main text we incorrectly stated the initial screening of nucleophiles was performed in D₂O. The screen was done in CDCl₃, as originally indicated in Table S2.
- We used ethyl groups and not methyl groups as the substituents in compounds **2a**, **2b** and **2e**. This was a typo (the correct compounds were used for stoichiometry calculations etc) and we have changed this throughout.
- In Figure 4, one of the signals was accidentally misassigned (proton a). The assignment was correct in all other places and no data is affected. The correct figure has now been added as replaced Figure 4.